# Effects of Obesogenic Feeding and Free Fatty Acids on Circadian Secretion of Metabolic Hormones: Implications for the Development of Type 2 Diabetes

**DOI:** 10.3390/cells10092297

**Published:** 2021-09-03

**Authors:** Alexandre Martchenko, Patricia Lee Brubaker

**Affiliations:** 1Department of Physiology, University of Toronto, Toronto, ON M5S 1A8, Canada; alexandre.martchenko@mail.utoronto.ca; 2Department of Medicine, University of Toronto, Toronto, ON M5S 1A8, Canada

**Keywords:** beta cell, GLP-1, high-fat diet, insulin, L cell, palmitate, western diet

## Abstract

Circadian rhythms are 24-h internal biological rhythms within organisms that govern virtually all aspects of physiology. Interestingly, metabolic tissues have been found to express cell-autonomous clocks that govern their rhythmic activity throughout the day. Disruption of normal circadian rhythmicity, as induced by environmental factors such as shift work, significantly increases the risk for the development of metabolic diseases, including type 2 diabetes and obesity. More recently, obesogenic feeding and its fatty acid components have also been shown to be potent disruptors of normal circadian biology. Two key hormones that are released in response to nutrient intake are the anti-diabetic incretin hormone glucagon-like peptide-1, from intestinal L cells, and insulin secreted by pancreatic β cells, both of which are required for the maintenance of metabolic homeostasis. This review will focus on the circadian function of the L and β cells and how both obesogenic feeding and the saturated fatty acid, palmitate, affect their circadian clock and function. Following introduction of the core biological clock and the hierarchical organization of the mammalian circadian system, the circadian regulation of normal L and β cell function and the importance of GLP-1 and insulin in establishing metabolic control are discussed. The central focus of the review then considers the circadian-disrupting effects of obesogenic feeding and palmitate exposure in L and β cells, while providing insight into the potential causative role in the development of metabolic disease.

## 1. Introduction

Type 2 diabetes (T2D) is a metabolic disease that, ultimately, results from insufficient insulin secretion to compensate for peripheral insulin resistance [1]. The main risk factor implicated in the global rise in T2D prevalence is the concurrent obesity epidemic [1]. Obesity is also a multi-faceted disease, characterized by increased adiposity and body weight as a result of an imbalance between energy intake and energy expenditure [2]. Although both genetic and epigenetic factors can be causative in obesity [2], the modern environment, with increased availability of low-cost, calorie-dense fast foods and a sedentary lifestyle, serves as a major contributor to the development of obesity. Interestingly, in recent decades, disruption of our endogenous circadian rhythms has emerged as another potential contributor to the increased prevalence of metabolic disease [3]. With the advent of genome-wide association studies, several polymorphisms in circadian clock and clock-controlled genes have been discovered to be associated with increased susceptibility to metabolic diseases including both T2D and obesity [4,5,6,7,8,9,10,11]. Furthermore, human epidemiological studies have linked circadian disruption, such as that induced by shift-work or increased exposure to artificial light at night, to increased rates of metabolic disease [12,13,14,15,16,17,18]. In parallel with the human data, an abundance of work conducted using both genetic knockout approaches and rodent models of light-induced circadian disruption has demonstrated a mechanistic link between circadian rhythms and metabolism [19,20,21,22,23,24,25]. However, over the last few years, in vivo studies as well as in vitro evidence also points to obesogenic diets and their components as disruptors of behavioral and metabolic circadian rhythmicity [26,27,28,29,30,31,32,33,34,35,36]. Although multiple circadian-related factors may synergize to promote disease, the major focus of this review will be the impact of fatty acids on the circadian function of two key metabolic cell types, the intestinal L cell that secretes glucagon-like peptide-1 (GLP-1), and the pancreatic β cell that releases insulin, as well as the mechanisms by which obesogenic diets may, ultimately, lead to metabolic dysfunction.

## 2. Circadian Rhythms

Circadian rhythms, which are present in virtually all known organisms, evolved as a result of constant predefined environmental cues, mainly the light–dark cycle [3]. This internal biological clock, with a period of approximately 24 h, allows organisms to have anticipatory physiological responses, thereby improving their biological fitness [37]. All nucleated cells of the body express what are termed “clock genes”, which are ultimately responsible for generating circadian rhythmicity within organisms [38]. At the fundamental level, the mammalian circadian clock machinery consists of two basic helix-loop-helix transcription factors, circadian locomotor output cycles kaput (CLOCK; encoded by *CLOCK*) and brain and muscle aryl hydrocarbon receptor nuclear translocator-like protein 1 (BMAL1; encoded by *ARNTL*), which heterodimerize and activate the *PERIOD* 1, 2, and 3 (*PER1/2/3*) and *Cryptochrome* 1 and 2 (*CRY1/2*) genes through binding to E-boxes in their promoters [38]. CRY and PER form a complex which then inhibits BMAL1/CLOCK, thereby forming an auto-regulatory transcriptional/translational feedback loop with an approximately 24-h period which is defined by the rate of PER and CRY degradation by casein kinases 1δ and -ε (CK1δ/ε) and the F-box/leucine rich-repeat protein 3 (FBXL3), respectively [39]. In addition to the core molecular loop, the BMAL1/CLOCK heterodimer also induces expression of the nuclear receptors, *REV-ERB**α* and *-**β* which act to repress *ARNTL* expression, as well as of retinoic acid related-orphan receptor α and γ (*ROR**α* and *-**γ*) which oppose REV-ERBα/β and stimulate *ARNTL* expression [40,41]. Importantly, these clock genes are also involved in transcriptional regulation of downstream clock-controlled genes, thereby establishing a circadian rhythm to key cellular and metabolic processes (Figure 1). It has been estimated that approximately 10–15% of the genome is under clock gene regulation within a given tissue while, as a whole within an organism, approximately 43% of all protein-coding genes display a circadian rhythm in their expression [42].

The molecular circadian clock was originally discovered in the hypothalamus and, specifically, within the suprachiasmatic nuclei (SCN) [43]. The SCN, which is often referred to as the “master clock”, is entrained by light and is comprised of two nuclei, each composed of approximately 10,000 neurons [43]. Each neuron expresses its own cell-autonomous rhythm; however, although the period of a single cell can range from 22 to 30 h, as a whole through intercellular communication, the circadian period is set to a fairly precise 24 h as entrained by the main zeitgeber or time-keeper, which is light exposure [43]. Coordination of downstream, peripheral tissue rhythmicity by the SCN is thought to be carried out, in part, through hormonal, temperature, and behavioral outputs, as well as through innervation of both the sympathetic and parasympathetic arms of the autonomic nervous system [43]. However, a key pathway by which the master clock synchronizes peripheral clocks, and which is crucial to the maintenance of metabolic homeostasis, is through regulation of the sleep/wake cycle that, ultimately, determines the fasting/feeding cycle and subsequent coordination of nutrient handling [44].

The essential nature of the SCN in regulating the endogenous clock was originally demonstrated in rats by ablating and subsequently re-engrafting the SCN, therein abolishing and restoring circadian rhythmicity, respectively [45,46,47]. However, animals with a lesioned SCN exposed to a regular feeding schedule exhibited anticipatory behaviors towards food, providing evidence for the presence of food-entrainable oscillators that act independently of the SCN [48]. In line with evidence for the ability of nutrient intake to act as a zeitgeber, metabolic tissues such as the gastrointestinal tract, pancreatic islet cells, liver, adipose tissue, and skeletal muscle have all been shown to exhibit their own, cell-autonomous circadian rhythmicity [21,49,50,51,52,53,54,55]. In combination, the circadian activity of these tissues is thought to be largely responsible for the coordination of metabolic homeostasis throughout the 24-h fasting/feeding cycle (Figure 2) [56].

## 3. Circadian Regulation of Metabolic Function

After a period of fasting, the gastrointestinal (GI) tract is the first metabolic tissue to come in contact with ingested nutrients. Therein, crucial functions, such as nutrient digestion and absorption, as well as enteroendocrine hormone secretion are all rhythmic processes that are driven by circadian clock gene expression within the gut. The circadian activity of the GI tract and its role in maintaining metabolic homeostasis has been recently been reviewed by Martchenko and colleagues [57]. However, one GI cell type that is central to this review is the enteroendocrine L cell, which makes up approximately 0.5% of all cells within the intestinal epithelium and is predominantly localized to the ileum and colon [58]. L cells secrete several peptide hormones including, most notably, the glucagon-like peptides (GLP), GLP-1 and GLP-2 [59]. Due to its anti-diabetic and satiety-inducing properties, GLP-1-derivative drugs have been developed and approved for the treatment of T2D and obesity [60]. In contrast, GLP-2 enhances intestinal growth as well as nutrient digestion, absorption and barrier integrity [61], and GLP-2-based therapeutics are currently used to treat patients with intestinal failure consequent to short bowel syndrome [61].

The intestinal L cell has recently been demonstrated to exhibit circadian activity. As determined by analysis of plasma GLP-1 levels, L cell secretion has been shown to differ by time-of-day in humans, with varying results produced by several different groups, likely as a result of differences in fasting times as well as nutrient loads [62,63,64,65]. Nonetheless, circadian disruptors such as phase-shifts, as well as exposure to light during the normal rest period have been shown to disrupt the rhythms in GLP-1 release [63,66]. The use of rat and mouse models, which allows for identical fasting times as well as the delivery of a standard oral glucose load, showed that GLP-1 secretion follows a 24-h rhythm with peak secretion occurring at the onset of the normal active/feeding period, consistent with the role of nutrient intake as a zeitgeber for L cell rhythmicity [33,67,68]. Furthermore, analysis of primary mouse L cells, as well as of both murine (m) GLUTag and human NCI-H716 L cell lines has revealed the existence of a cell-autonomous circadian clock, as evidenced by anti-phasic expression of the positive and negative arms of the core molecular clock (i.e., *Arntl* and *Per2*, respectively) [33,67,68].

The metabolic relevance of circadian GLP-1 secretion is largely thought to be its role in entraining circadian insulin secretion. Hence, more pronounced rhythms in insulin release are observed following oral rather than intravenous nutrient administration, consistent with the release of GLP-1 only following nutrient intake [69]. Furthermore, identical doses of GLP-1 in combination with the same glucose load result in differential insulin responses based on the time of day in both rat and mouse models [33,67]. GLP-1 receptor agonists have also been shown to synchronize rhythmic activity of pancreatic β cells in vitro [70]. In contrast to GLP-1, much less is known about the role of circadian GLP-2 secretion. However, given its actions stimulating the digestion and absorption of nutrients, including fatty acids [61], rhythmic release of GLP-2 is presumed to also contribute to the circadian control of metabolic homeostasis. Finally, the diurnal activity of the intestinal L cell has recently been shown to be dependent upon the intestinal microbiome [33] which, itself, demonstrates a circadian rhythm related to the fasting/feeding cycle [71,72]. Although specific microbial metabolites that regulate circadian L cell secretion have not been identified, short chain fatty acids and secondary bile acids also demonstrate circadian patterns [73,74] and are known L cell secretagogues [75,76]. In addition, several molecular pathways that contribute to the rhythmic release of the GLP-containing secretory granules have been elucidated, most notably as related to the SNARE exocytotic proteins, syntaxin-binding protein-1 and secretagogin (Figure 3) [68,77]. Although current GLP-1-based therapies for T2D and obesity are long-lasting, future work should focus on developing secretagogue-based therapies that will take advantage of the natural GLP-1 secretory rhythm.

The pancreatic islet contains three main cell types: the glucagon secreting α cells, insulin secreting β cells, and somatostatin releasing δ cells. Little is known about circadian rhythms in the δ cells. However, both the α and β cells display circadian gene expression profiles which drive rhythmic insulin and glucagon secretion, respectively [51,52,70,78]. Of note, the α cells are also known to release GLP-1, which would also presumably follow a rhythmic pattern but, due to minimal amounts entering the circulation, are unlikely to be of great importance for maintaining metabolic homeostasis [79,80]. Interestingly, the circadian rhythmicity of pancreatic α and β cells are distinct, with the core molecular clock of the β cells exhibiting an approximately 4-h phase-advance in relation to the α cells, likely to ensure appropriate hormone secretion during the feeding and fasting periods, respectively [52]. Importantly, transcriptomic analysis of isolated α and β cells has identified rhythmicity in genes involved in key islet pathways such as glucose sensing and processing, as well as granule transport and exocytosis in both cell types [51,52,78]. Similar to the intestinal L cell, several exocytotic SNARE and SNARE-accessory proteins which are essential to islet hormone secretion have also been identified as exhibiting circadian expression, including vesicle-associated membrane proteins, syntaxins, syntaxin-binding proteins and synaptotagmins, as well as secretagogin (Figure 3) [51,52,78].

The essential role of the β cell clock in metabolic homeostasis has been extensively studied. Pancreas-specific *Arntl* knockout (KO) mice, which maintain normal circadian activity and feeding patterns, were found to be hyperglycemic due to impaired insulin secretion [21,81]. Ex vivo studies revealed impairments in insulin granule exocytosis in KO mice, while morphological analysis of the pancreas demonstrated reductions in β cell proliferation [21,81]. Further studies in β cell-specific *Arntl* KO mice revealed the importance of the core clock in ensuring proper β cell maturation, as characterized by alterations in gene expression patterns and glucose-stimulated insulin secretion (GSIS) [78,82]. Furthermore, β cell-specific *Arntl* overexpression results in enhancement of the circadian amplitude and improved GSIS while protecting against obesity-induced glucose intolerance [83]. These finding are consistent with a report demonstrating that *Arntl* expression is induced in the β cell during post-natal maturation [82]. Collectively, therefore, the temporal response to feeding is initiated by nutrient ingestion, gastrointestinal digestion, absorption and secretion of hormones such as GLP-1 and GLP-2, and subsequent release of insulin and suppression of glucagon. This process optimizes nutrient deposition during the feeding period into metabolic tissues that are regulated in parallel not only by insulin but, also, by their own metabolic clocks in the liver, skeletal muscle and adipose tissue, as recently reviewed [84,85,86].

## 4. Effects of Obesogenic Feeding and Free Fatty Acid Exposure on Circadian Secretion of L Cell and β Cell Hormones

In a seminal study on the relationship between obesogenic feeding and circadian rhythms, Kohsaka et al. showed that ingestion of a high-fat diet (HFD; 45% total calories from fat) disrupts both locomotor and feeding patterns, such that the mice have increased activity and nutrient intake during their normal light or rest period [26]. Interestingly, the effects on circadian period, locomotion, and food intake are observed following just one week of HFD-feeding and prior to any significant body weight gain, implicating the nutrients themselves in the circadian disruption [26]. Another model of obesogenic feeding, the high-fat (41% of total calories)/high-sucrose (29% of total calories) Western diet (WD) has also been examined for its role as a disruptor of endogenous circadian rhythmicity. Similar to the HFD studies, WD feeding in mice disrupts food intake, shifting the overall pattern towards consumption of more calories during the rest period [28,33]. Interestingly, neither HFD- nor WD-feeding seemed to affect clock gene mRNA expression when the whole hypothalamus was analyzed [26,28]. However, select studies have identified disruption in *Per2* mRNA and PER2 protein expression in several brain regions following high-fat/high-sugar feeding in mice including the epithalamic lateral habenula as well as the reward-related nucleus accumbens; however, no changes were observed in the SCN or the arcuate nucleus (ARC) [87,88]. The ARC contains the orexigenic neuropeptide Y/agouti-related peptide neurons as well as the anorexigenic proopiomelanocortin neurons, which are key regulators of nutrient intake [89]. The lack of marked effects of obesogenic feeding on the core molecular circadian clock within the ARC of the hypothalamus may suggest that the effects on food intake patterns are independent of the circadian clock machinery. However, although obesogenic feeding may not disrupt the central circadian clock, HFD, WD and specific fatty acids have been shown to disrupt circadian rhythms in both L and β cells.

### 4.1. Obesogenic Diets

The effects of obesogenic feeding on circadian release of GLP-1 (and thus, of GLP-2) from the intestinal L cell have been well studied in both humans and rodent models. Human studies have shown that obese individuals lose their GLP-1 secretory rhythm [64], while morbidly obese individuals with T2D have altered GLP-1 secretory rhythms as compared to those with a similar body weight but normal glucose tolerance [65]. Similarly, rats fed a WD for five weeks lose their normal pattern in GLP-1 release, such that secretion during the normal trough period (i.e., at the onset of the inactive/fasting period) is elevated [31]. More extensive analyses subsequently conducted using mice fed a WD for 16 weeks demonstrated that these animals completely lose their rhythm in GLP-1 release in association with massive elevations in GLP-1 across the 24-h day; transcriptomic analysis of their primary L cells also showed a disruption in their circadian clock with WD-feeding [33]. Notably, the sustained increase in GLP-1 levels in the WD-animals was vital to the maintenance of oral glucose homeostasis throughout the day/night cycle in these animals, as similar effects were not seen following intravenous glucose tolerance testing [33]. Furthermore, the elevations in GLP-1 were determined to be dependent on the intestinal microbiome [33]. While this was correlated to an increase in the abundance of *Akkermansia muciniphila* [33], a species that has been extensively linked to GLP-1 release [90,91], the underlying mechanisms are yet to be defined. Indeed, as these studies were conducted using a WD, which is high in both fat and sucrose, while being low in fiber, it is not possible to define a role for specific nutrients in the L cell circadian clock. Analysis of the circadian function of the primary L cell under fiber-controlled, high-fat feeding conditions is therefore warranted.

Studies in rodent models on HFD (both 45% and 60% of total calories from fat dietary compositions)-feeding have also demonstrated disrupted clock gene expression and hormone secretion by the pancreatic islets [27,92]. Obesogenic feeding leads to several metabolic adaptations within the body, which include an increase in pancreatic β cell mass and function to compensate for the HFD/WD-induced insulin resistance, thus preventing or delaying the onset of T2D. The importance of the circadian clock to this process was made by the demonstration that β cell-specific *Arntl* KO mice fail to expand their β cell mass and function in response to HFD-feeding [92]. Further work in rats has also suggested that progression to diabetes is characterized by synergism between circadian disruption (as induced by constant-light exposure) and ingestion of a HFD [27]. Interestingly, these changes were attenuated by co-treatment with either metformin or melatonin, which both decrease hepatic glucose production and attenuate the circadian disruption, although the underlying mechanisms are not entirely clear [29].

### 4.2. Fatty Acids

The saturated fatty acid palmitate is a major component of both Western and high-fat diets, and is the most abundant fatty acid in the plasma upon obesogenic feeding [93]. As a consequence, palmitate has been extensively studied for its effects on L and β cell function and, more recently, for its circadian-disrupting effects in these metabolic cell types. Although, in the acute setting, palmitate does not modulate GLP-1 secretion [94,95], exposure to palmitate has been shown to disrupt normal L cell physiology. Palmitate treatment of mGLUTag L cells induces ER stress, as characterized by increased expression of C/EBP homologous protein (*CHOP*) and binding immunoglobulin protein (*BIP*), as well as by decreased expression of prohormone convertase 1/3 (*Psck1*) and reduced GLP-1 content and secretion [31,32,96]. Exposure to palmitate also increases reactive oxygen species in mGLUTag L cells in association with elevated caspase-3 activation, although cell viability and apoptosis were not affected in this study [97]. Similar to the L cell, pancreatic β cell function is also diminished by exposure to palmitate (Figure 3). Pancreatic islets treated with palmitate exhibited increased levels of reactive oxygen species in association with impaired mitochondrial function and, ultimately, decreased GSIS [98,99,100]. Furthermore, palmitate also induces ER stress and increases apoptosis in the β cell, although the mechanisms are not yet fully understood [98,99]. Interestingly, exposure of pancreatic β cells to palmitate has also been shown to disrupt the physiologic effects of GLP-1 signaling, thereby decreasing the subsequent insulin secretory response [101].

In addition to dampening function, palmitate has been shown to exert circadian-disrupting effects in the L cell. Using the mGLUTag L cell line, it was first determined that exposure to palmitate dampens the amplitude of *Arntl* expression as well as disrupting the normal period and amplitude of *Per2* expression [31]. Furthermore, treatment of mGLUTag L cells with palmitate decreases the expression of key downstream, clock-dependent genes, including nicotinamide phosphoribosyltransferase (*Nampt*) and sirtuin 1 (*Sirt1*) [32]. The enzyme NAMPT catalyzes the key rate-limiting step in the synthesis of the co-enzyme, nicotinamide adenine dinucleotide (NAD^+^) from its precursor nicotinamide [102,103]. NAD^+^ is crucial in determining cellular energy status, at least in part, through the sirtuin NAD^+^-dependent deacetylase proteins, with SIRT1 localized mainly in the nucleus and SIRT3 in the mitochondria [102,103]. Collectively, within the mGLUTag L cells, palmitate-induced disruption of the L cell clock suppresses NAMPT activity, reduces NAD^+^ levels and impairs mitochondrial function and ATP production, resulting in impaired time-dependent GLP-1 secretion (Figure 3) [32]. These findings are consistent with similar palmitate-induced clock-disrupting effects in other metabolic cell types, including hypothalamic neurons, hepatocytes, and adipocytes [34,35,36,104]. Interestingly, studies in hepatocytes have shown that, in this tissue, palmitate does not alter the expression of BMAL1 and CLOCK but, rather, destabilizes their interaction by inhibiting the activity of SIRT1 [34,35]. In accordance with these data, palmitate interferes with the deacetylation of BMAL1 by SIRT1, and reduces the transcriptional activity of the BMAL1/CLOCK heterodimer [35]. However, further work needs to be conducted in the L cell to determine the entirety of the mechanism(s) underlying palmitate-induced disruption of the circadian function in this metabolic cell type.

Although the detrimental effects of palmitate are well-established in terms of decreasing normal pancreatic β cell function [99], few studies have focused on the effects of this saturated fat on the circadian activity of the β cells. One study using normal human islets treated with palmitate identified a disruption in expression of the core clock genes *Per1* and *Per3*, but no further downstream mechanisms were identified [105]. Interestingly, activation of SIRT1 by resveratrol rescues palmitate-induced β cell dysfunction, suggestive of a clock-dependent mechanism [106,107]; however, more mechanistic studies are required. Finally, islets from patients with T2D have been shown to have altered clock gene expression patterns in association with disrupted insulin release [108,109]. Although it remains unclear whether β cell clock disruption is a cause or consequence of T2D, treating such islets with nobiletin, a RORα agonist, has positive effects on both clock gene expression and insulin release [83,108,109]. Overall, although obesogenic feeding and palmitate exposure are established disruptors of β cell circadian rhythms, future work should focus on understanding the potential causative role for impairment of the β cell circadian clock in T2D.

## 5. Conclusions

Circadian activities of key metabolic tissues function together to coordinate nutrient intake, digestion, absorption, and utilization throughout the 24-h day, which is critical to the maintenance of metabolic homeostasis. To date, numerous studies have outlined the detrimental effects of obesogenic feeding, as well as of specific free fatty acid components, mainly the saturated fatty acid palmitate, on circadian rhythmicity in metabolic tissues. Based on current evidence, the deleterious effects of increased fat consumption can precede significant body weight gain, suggesting that specific fatty acids are causative in the disruption of metabolic tissue circadian rhythms and, in particular, those of the intestinal L cell and the pancreatic β cell. Given the essential roles of both GLP-1 and insulin in maintaining metabolic homeostasis, it is therefore clear how altered circadian function of the L and β cells may, thus, pre-dispose to the progression of diseases such as T2D. However, future research is required to elucidate whether diet-induced circadian disruption within these metabolic tissues is causative or merely associated with metabolic disease. Furthermore, it would be interesting to elucidate which tissue clocks are affected first and whether some of these are more protected against the stresses of obesogenic feeding, thereby providing a more complete picture of the timeline in disease development.

## Figures and Tables

**Figure 1 cells-10-02297-f001:**
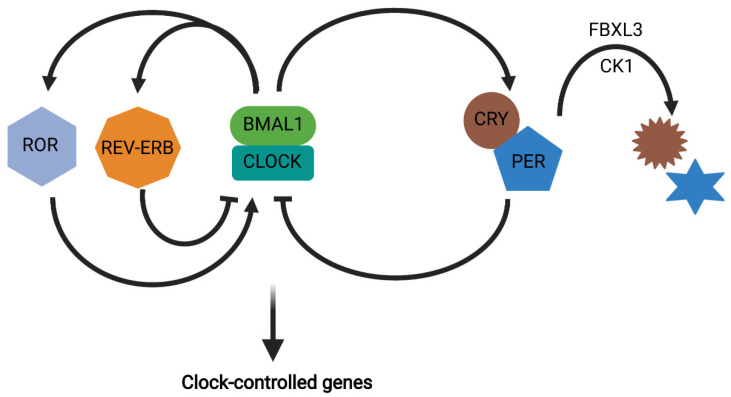
The core molecular clock consists of a positive arm in which BMAL1 and CLOCK heterodimerize to induce expression of the negative arm consisting of PER and CRY, which then feedback to inhibit BMAL1 and CLOCK. Additionally, BMAL1/CLOCK activate REV-ERBα/β and RORα/γ which have opposing roles in regulation of *ARNTL* (which encodes BMAL1), whereby REV-ERBα/β stimulates and RORα/γ inhibits *ARNTL* expression. Protein degradation mediated by CK1δ/ε and FBXL3, generates the 24-h circadian period. Finally, transcriptional regulation of clock-controlled genes by the core clock establishes a circadian rhythm in a wide range of cellular functions. Figure created with BioRender.com.

**Figure 2 cells-10-02297-f002:**
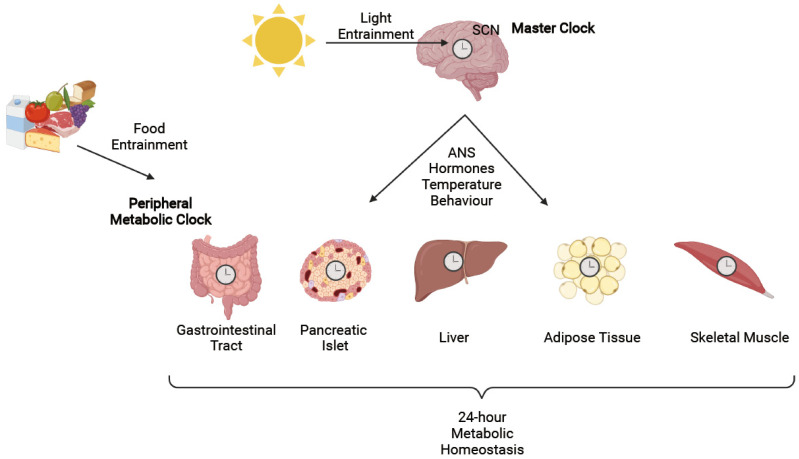
Light, the strongest zeitgeber, entrains the SCN which is the master clock. The SCN sends out signals through the autonomic nervous system (ANS), as well as hormonal, temperature, and behavioral outputs to synchronize peripheral tissues. However, food intake can directly synchronize peripheral metabolic clocks through the presence of food entrainable oscillators that are independent of the SCN. Collectively, the peripheral metabolic clock is responsible for establishing diurnal metabolic homeostasis. Figure created with BioRender.com.

**Figure 3 cells-10-02297-f003:**
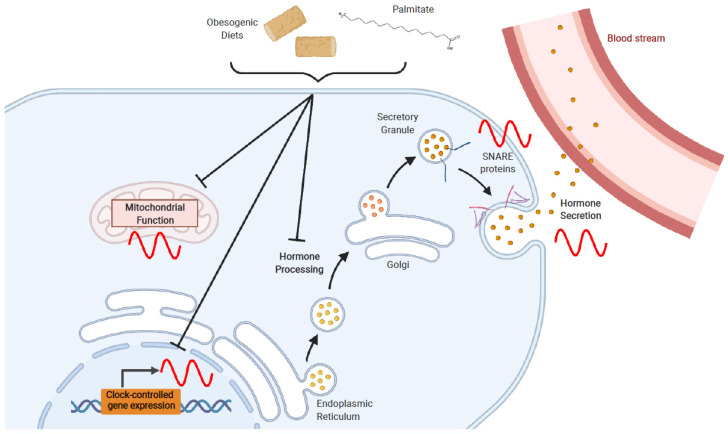
Circadian clock driven expression of key cellular pathways establishes rhythmic hormone release from the intestinal L cell and the pancreatic β cell. Ultimately, circadian secretion results from the rhythmic expression of key SNARE proteins involved in exocytosis. The obesogenic diet and its major component, the saturated fatty acid palmitate, have been shown to disrupt key L and β cell processes including clock and clock-controlled gene expression, mitochondrial function, and hormone processing through induction of ER stress. Collectively, exposure to high fat results in impaired rhythmic hormone secretion from both of these endocrine cell types. Figure created with BioRender.com.

## Data Availability

Not applicable.

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
