# Peer review of "Effects of Obesogenic Feeding and Free Fatty Acids on Circadian Secretion of Metabolic Hormones: Implications for the Development of Type 2 Diabetes"

_cells, 2021, doi:10.3390/cells10092297_

Round 1
Reviewer 1 Report
In this study, the authors reviewed the circadian function of L-cells and beta-cells and the effects of obesity-fed and saturated fatty acid palmitate on circadian rhythm and function. Please find specific suggestion below.
Please add animations or tables representing the contents of each section (2.Circadian rhythms, 3. Circadian Regulation of Metabolic Function. 4. Effects of Obesogenic Feeding and Free Fatty Acid Exposure on Circadian Secretion of L cell and β cell Hormones (Obesogenic Diets & Fatty Acids))
Reviewer 2 Report
The current review focuses on the circadian function of the L and beta-cells and how both obesogenic feeding and the saturated fatty acid, palmitate, affect their circadian clock and function.
The topic is really interesting and new, however the manuscript appears confused and difficult to read. The Authors should strive to make the article more linear and should provide a clue to potential clinical applications.
Comments:
Paragraph 2 should be enriched with an explanatory figure.
A graph representing and elucidating the rhythmic secretion of GLP-1 should be included in section 3.
Given the rhythmicity of GLP-1 function, GLP-1 receptor agonists should be administered preferably at a specific time of day?
GLP-1 is also expressed by alpha-cell. In this case too, is the secretion of GLP-1 rhythmic?
Page 6, lines 265-268: it should be added that palmitate is also able to disrupt the physiological effects of GLP-1 at beta-cell level (Endocrinology. 2016 Jun;157(6):2243-58. doi: 10.1210/en.2015-2003).
Paragraph 4 should be completed with 2 or more resuming tables.
How the rhythmicity of GLP-1 and insulin could influence each other?
After reading the paper, it is not clear why GLP-1 and insulin should be considered rhythmic hormones, as well as the fact that they are secreted after the ingestion of nutrients.
Reviewer 3 Report
The authors have provided a well organized, well written, important review of the literature relating to the impact of high fat on circadian control of metabolic hormones. The emphasis on the roles of L and ß cells, and the disruption of circadian functions in these cells by high fat, is particularly interesting. Some minor revisions are suggested, and adding a schematic figure might add to the impact of the review:
1) Could a figure/schematic be provided to highlight the major points of the review? For example, an image showing the key impacts of high fat on L cell and ß cell circadian secretion of hormones would be helpful to put the major concepts of the review in visual context.
2) There are some locations in the text where L and ß cells are referred to as tissues, rather than cells (e.g. line 53). This might need to be checked.
3) Line 58 indicates that circadian rhythms are present in all organisms. Should soften this statement for accuracy.
4) Line 67 should read "heterodimerize and activate".
5) Lines 213-214: readability would be improved by changing to "However, although obesogenic feeding may not disrupt the central circadian clock, HFD, WD and specific fatty acids have been shown to disrupt circadian rhythms in L and ß cells."
6) Lines 303-305: to improve readability, could change to "Overall, although obesogenic feeding and palmitate exposure are established disruptors of ß cell circadian rhythms, future work should focus on understanding the potential causative role for impairment of the ß cell circadian clock in T2D."
Round 2
Reviewer 2 Report
The Authors have added 3 identical figures. I think this is an error. Please, add 3 different figures.
Round 3
Reviewer 2 Report
The Authors have addressed all my requesrs.